



# Learning-based prediction of the particles catchment area of deep ocean sediment traps

Théo Picard [1], Jonathan Gula [2,3], Ronan Fablet [4], Jeremy Collin [1], and Laurent Mémery [1]

[1]Univ Brest, CNRS, IRD, Ifremer, Laboratoire des Sciences de l'Environnement Marin (LEMAR), IUEM, Plouzané, France
[2]Univ Brest, CNRS, IRD, Ifremer, Laboratoire d'Océanographie Physique et Spatiale (LOPS), IUEM, Plouzané, France
[3]Institut Universitaire de France (IUF), Paris, France
[4]IMT Atlantique, Lab-STICC, Plouzané, France

**Correspondence:** Théo Picard (theo.picard@univ-brest.fr)

**Abstract.** The ocean biological carbon pump plays a major role in climate and biogeochemical cycles. Photosynthesis at the surface produces particles that are exported to the deep ocean by gravity. Sediment traps, which measure the deep carbon fluxes, help to quantify the carbon stored by this process. However, it is challenging to precisely identify the surface origin of particles trapped thousands of meters deep because of the influence of ocean circulation on the carbon sinking path. In this study, we conducted a series of numerical Lagrangian experiments in the Porcupine Abyssal Plain region of the North Atlantic and developed a machine learning approach to predict the surface origin of particles trapped in a deep sediment trap. Our numerical experiments support its predictive performance, and surface conditions appear to be sufficient to accurately predict the source area, suggesting a potential application with satellite data. We also identify potential factors that affect the prediction efficiency and we show that the best predictions are associated with low kinetic energy and the presence of mesoscale eddies above the trap. This new tool could provide a better link between satellite-derived sea surface observations and deep sediment trap measurements, ultimately improving our understanding of the biological carbon pump mechanism.

## 1 Introduction

The Biological carbon pump (BCP) plays a major role in climate and biogeochemical cycles. The BCP reduces unperturbed atmospheric CO2 by 35% to 50% (Williams and Follows, 2011) by exporting organic matter to the deep ocean, thereby supporting abyssal food webs (Billett et al., 1983; Rembauville et al., 2018). The BCP is driven by photosynthesis that occurs within the euphotic layer, typically between 0 and 200 meters, producing gravitationally sinking particulate organic carbon (POC). Sinking particles have a wide range of vertical velocities, from neutral buoyancy to more than 600 meters per day (Villa-Alfageme et al., 2016), and are usually considered to be the main contributors to the BCP (Armstrong et al., 2001; Alonso-González et al., 2010; Siegel et al., 2014; Le Moigne, 2019). While most of POC is remineralized in the euphotic and the mesopelagic zones (200 - 1000 meters), a small but significant fraction of POC reaches the deep ocean below 1000 m to be sequestered for hundreds or thousands of years (Lampitt et al., 2008; Burd et al., 2016). Despite its critical importance, the annual estimate of global carbon export remains poorly constrained, ranging from 5 to over 12 Pg C per year (Turner,



2015). Therefore, quantifying the biological carbon pump is key to understanding the global carbon cycle and how the BCP will respond to climate change (Passow and Carlson, 2012; Henson et al., 2022).

Export production has historically been observed with sediment traps (STs) from the upper thermocline down to several thousand meters (Le Moigne et al., 2013). However, linking local primary production to observed exported carbon fluxes is a challenging task. To effectively interpret ST time series, the catchment area, defined as the surface domain containing all likely positions from which particles entering the trap could originate (Deuser et al., 1988), must be clearly identified. Traditionally, the catchment area is considered to be the zone directly above the traps (Deuser and Ross, 1980; Alldredge and Gotschalk,

1988; Asper et al., 1992; Armstrong et al., 2001; Lampitt et al., 2023). While this approach demonstrates a first-order coupling, it is based on a strong hypothesis: the POC export by gravitational sinking is considered from a quasi one-dimensional (1D) perspective. However, the ocean circulation significantly affects the particle sinking paths. Thus, the origin of the particles that reach a deep ST can be distributed over a large domain that depends mainly on the ocean dynamics at all scales (Siegel et al., 1990; Deuser et al., 1990; Burd et al., 2010; Liu et al., 2018; Dever et al., 2021)

Recent studies have focused on the influence of ocean circulation and evaluated funnel statistics for specific sediment trap locations using particle backtracking and numerical simulations (Siegel et al., 2008; Liu et al., 2018; Wekerle et al., 2018; Wang et al., 2022). These studies have shown that the catchment area is highly dependent on several factors such as sinking velocities, trap depth, regional and seasonal advective processes. The application of Lagrangian backtracking approaches with assimilated ocean currents has been proposed to relate the ocean circulation to real ST observations (Frigstad et al., 2015; Ma

et al., 2021). However, the full 3D reconstruction of the ocean states from available satellite-derived and in situ observations is highly challenging and prone to significant uncertainties in the retrieval of mesoscale and submesoscale dynamics (Cutolo et al., 2022). This leads to uncertainties in the prediction of catchment areas and in the assessment of the BCP.

Machine learning tools are increasingly being used to tackle these complex problems, and generally offer an end-to-end formulation that is easier to develop and computationally cheaper. Oceanography studies have already demonstrated the bene-

fits of machine learning, particularly in predicting the properties of the ocean interior from observations of the ocean surface (Chapman and Charantonis, 2017; Bolton and Zanna, 2019; George et al., 2021; Manucharyan et al., 2021; Pauthenet et al., 2022). They can achieve state-of-the-art performance or even outperform standard operational interpolation data assimilation schemes when focused on specific ocean variables (Manucharyan et al., 2021; Beauchamp et al., 2022; Cutolo et al., 2022). They can also significantly reduce the computational complexity of numerical simulations, including for Lagrangian parti-

cle trajectories (Jenkins et al., 2022). It is therefore clear that machine learning offers a wide range of new possibilities for improving our understanding of the ocean.

In this study, we use deep learning schemes to study the catchment areas of deep ocean particles. Our main contributions are as follows: (i) we formulate the prediction of the catchment area of particles trapped in STs as the supervised learning of a regression model from ocean circulation variables; (ii) we investigate whether ocean surface conditions constrain sinking

particles path; (iii) we analyse the main factors affecting the prediction performance. We report numerical experiments for a case study area in the North Atlantic using realistic high-resolution simulation data.



The paper is organized as follows. Section 2 introduces the methodology use to design the databases for training and performance evaluation. Section 3 presents the neural network structure. In section 4, we evaluate the accuracy of the predictions, compare different configurations and analyse the relationship between input conditions and prediction performance. In Section
5, we discuss potential improvements and future work. Our conclusions are presented in Section 6.

## 2    Data and case-study area

As a case study, we focus on the origin of particles captured by moored ST at the UK Porcupine Abyssal Plain (PAP) station (49°N, 16.5 °W), which provides fluxes with a temporal coverage of more than 30 years (Hartman et al., 2021). We use a simulation-based experimental setup consisting of realistic CROCO simulations and particle backtracking experiments. More
precisely, we simulate particles falling into a fictive PAP ST located at 1000 m depth, focusing on particles with a vertical sinking velocity of 50 m/day.

### 2.1    Numerical simulation around PAP station

We use the realistic North Atlantic Subpolar Gyre simulation designed and validated by Le Corre et al. (2020). This simulation is run using CROCO (Coastal and Regional Ocean COmmunity model) based on ROMS (Shchepetkin and McWilliams, 2005),
which solves the hydrostatic primitive equations for the momentum and state variables. The domain has $2000 \times 1600$ grid points and a horizontal grid resolution of 2 km, resolving the mesoscale and part of the submesoscale. The model has 80 vertical sigma levels, with a vertical spacing of $\sim 5$ m at the surface and up to 100 m in the intermediate layer. After a two-year spin-up period, two simulations called POLGYR1 and POLGYR2 are run. POLGYR1 runs between Jan, 1th 2002 and Dec, 4th 2009 (eight years) and is used to create a training and an evaluation database, while POLGYR2 runs between Jul, 24th 2003 and Jan 12nd
2009 (about 6 years and half) and is used for the test database. The two simulation setups are identical in all aspects except for the initial and boundary conditions, which are perturbed in POLGYR2. After spin-up, the chaotic evolution results in uncorrelated dynamics between the two simulations (not shown here). We save snapshots every 12 hours for both simulations. We focus on a $1040 \times 1040$ km subdomain centered on PAP station (Figure 1). This region is characterized by moderate kinetic energy compared to the western and northern parts of the subpolar gyre, with a mean flow of about 0.05 m/s (Le Cann, 2005).

### 2.2    Lagrangian backtracking of catchment areas

A series of Lagrangian experiments are performed offline with Pyticles (Gula and Collin, 2021), using 3d velocities from the POLGYR1/2 simulations. Pyticles is a parallel Fortran/Python Lagrangian tool for offline 3D advection with the ability to include particle behaviour. Transport is performed along the native Arakawa C-grid and terrain-following vertical coordinates of the ocean model. The 3D velocity fields are linearly interpolated at particles positions, which are advected using a Runge-
Kutta 4 numerical scheme. In addition to passive advection, a negative vertical velocity is applied to simulate the sinking of dense particles. The numerical schemes have been shown to be robust to trajectory reversibility (Wang et al., 2022) using 12-hourly input fields and a time step of 120 seconds.





**Figure 1.** Snapshot of the North Atlantic Subpolar Gyre simulation on 8 February 2008 with relative vorticity. The dashed square is the domain of the simulation. The solid square is the sub-domain we are focusing on. The location of the PAP station is indicated by the black star. The black contour is the bathymetry at 1000 m depth.



For each experiment, we apply the following procedure. At 1000 m depth, we release in backtracking 36 particles every 12 hours over a period of 10 days (particle collection period), for a total of 720 particles. The particles are released uniformly over a patch of 10 km x 10 km and ascend until they reach the base of the euphotic layer at 200 m. Particles have a constant sinking velocity of 50 m/day and their journey takes on average 15 days, but this can vary from 10 to 20 days (Wang et al., 2022). We save the particles' positions every 100 m step and compute the resulting Probability Density Function (PDF). Figure 2a illustrates a typical experiment, in which most of the particles are attracted to an anticyclone structure visible between 200 m and 1000 m.Biological particles are mostly created between the surface and 200 m, but we considered as a first approach that the PDF at 200 m represents the depth from which they are exported. The PDFs saved at vertical levels lower than 200 m provide information on the 3D+t path of the particles to better constrain the training phase of the network. To improve training efficiency, we also apply a Gaussian filter to each PDF (Figure 2b, c). We performed sensitivity tests to ensure that increasing the number of particles and the size of the patch does not significantly affect the PDF. To avoid common particle between two experiments, we space them 10 days apart. Instead of representing one ST at PAP, we model 36 STs around the region to increase the number of experiments (Figure 3a ). The STs are close enough to the PAP site to have the same hydrodynamical properties. We choose to space the centres of the patches 36 km apart, so that particles from two different patches are separated by at least 26 km, which is slightly above the Rossby radius value in the region (Chelton et al., 1998). This distance is sufficient to observe significant differences in the catchment areas for two consecutive patches (Figure 3b).

For each Lagrangian experiment, we create an output tensor $Y_i = (n_{output} \times \delta x \times \delta y)$ , where $\delta x$ and $\delta y$ represent the number of points on the horizontal axis and $n_{output}$ the number of vertical levels where the PDFs are computed. In our case, we consider $n_{output} = 8$, corresponding to one layer every 100 m depth from 900 m to 200 m. The horizontal domain was chosen based on the statistical results presented in Wang et al. (2022). The backtracked particles are distributed within a 800 km x 800 km domain centered on PAP. Considering a horizontal resolution of 8 km (raw inputs are downscaled for storage constrain), this results in $\delta y = \delta x = 100$ points.

For each output $Y_i$, we store a corresponding input tensor $X_i = (n_{input} \times \delta x \times \delta y)$ which contains the hydrodynamic conditions of the experiment $i$. These data include temperature, vorticity, horizontal velocities (u and v), and sea surface height (SSH). All the variables are extracted on the same horizontal domain as $Y_i$ and downscaled to a resolution of 8 km. To capture the temporal variability, we consider a 10-day time sampling over a 30-day time window (4 time steps). To evaluate the importance of the surface in comparison to the sub-surface data (Section 4), we distinguish two datasets, $D_{4layers}$ and $D_{surf}$:

- $D_{4levels}$: this dataset contains the variables at 4 vertical levels, i.e. surface, 200 m, 500 m, 1000 m. Each input tensor is a $68 \times 100 \times 100$-dimensional tensor, where $n_{input} = 68$ represents the number of input fields extracted for each experiment.

- $D_{surf}$ : This dataset contains only sea surface conditions. This results in $20 \times 100 \times 100$-dimensional input tensors.





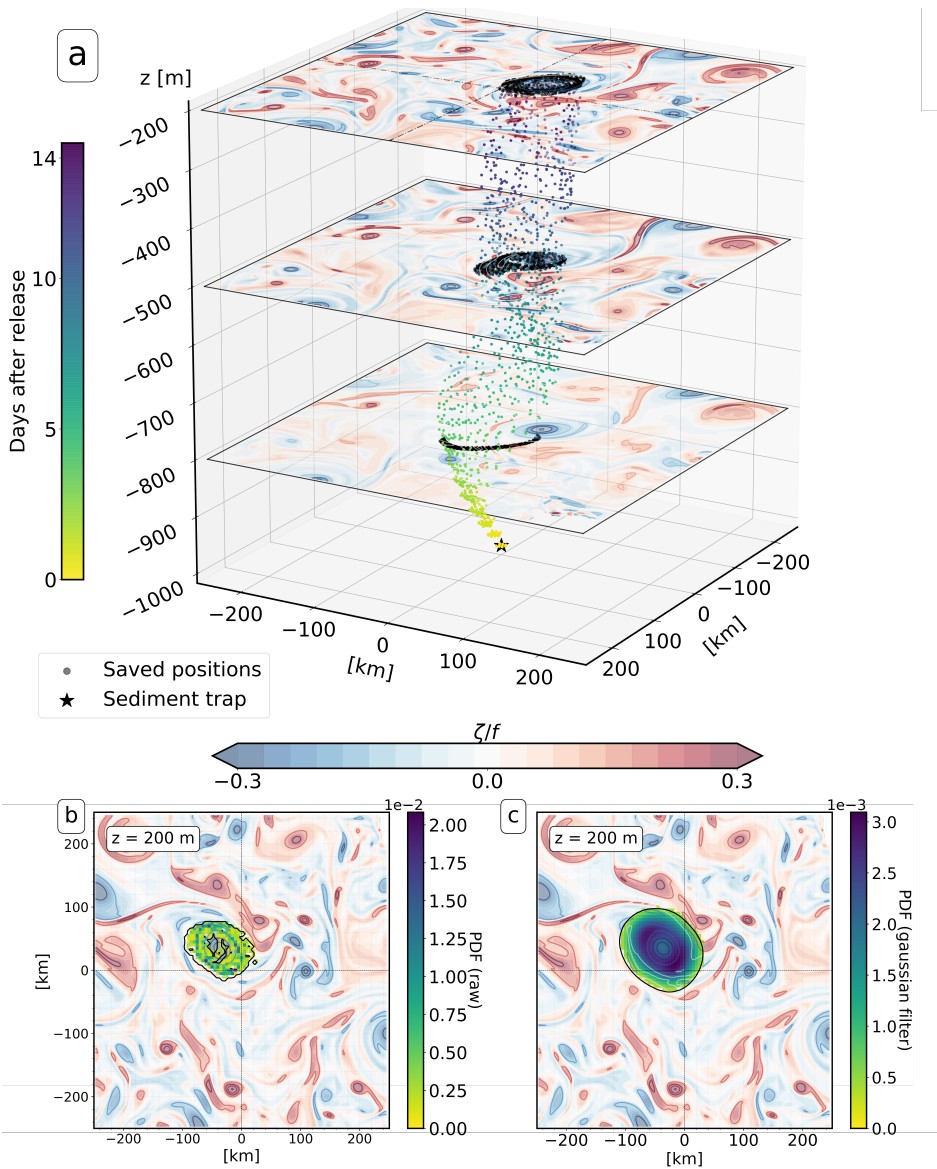

**Figure 2.** a) 3D view of the Lagrangian experiment. The color of the particles represents the number of days after their release at the ST (5% of particles are plotted). Snapshots of relative vorticity are displayed at 3 vertical layers (-800 m, -500 m, -200 m). Black points on the visualization indicate all the saved positions of particles at these specific vertical levels. b) The computed raw PDF of particles at 200 m c) PDF at 200 m after the Gaussian filter.





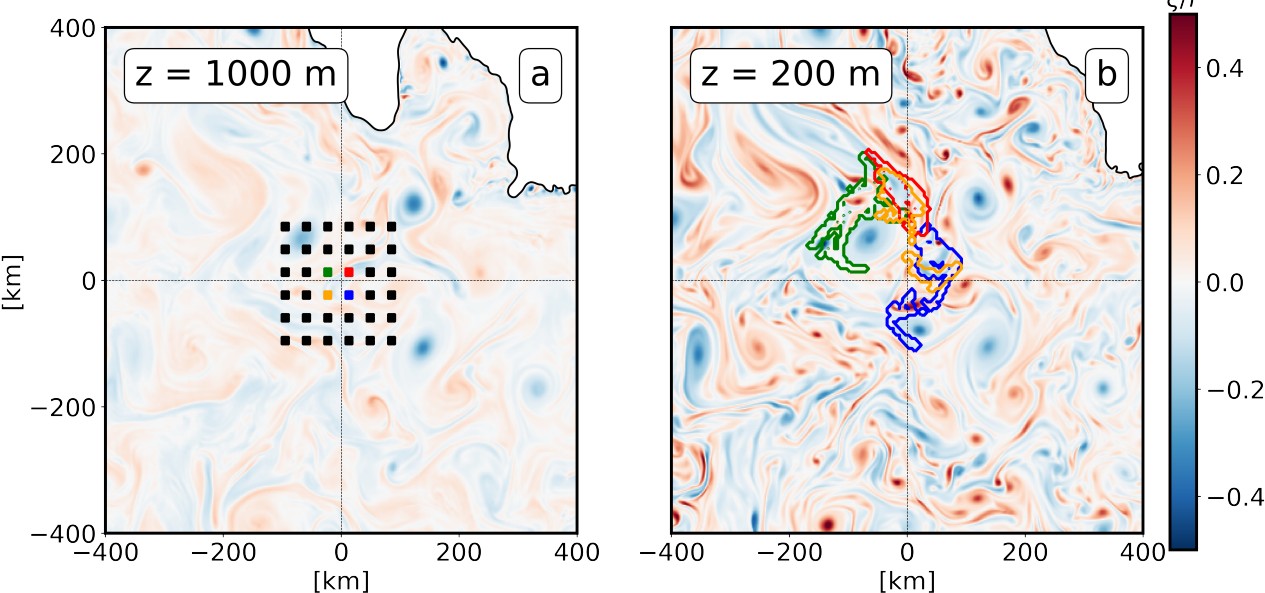

**Figure 3.** Snapshots of the relative vorticity on Oct 22 2003 (POLGYR1), 20 days after the particle release. a) vorticity field at a 1000 m depth with the 36 initial positions of the particles (black and colored squares). b) vorticity field at a 200 m depth with superimposed the origins of the backtracked particles for the 4 colored patches of the left panel.

## 3   Proposed deep learning scheme

120    This section introduces the proposed deep learning scheme, in particular the proposed neural architecture and the training scheme.

### 3.1   Training, validation and test data

Following the training and evaluation frameworks used in deep learning studies (Lecun et al., 2015), we consider independent training, validation and test datasets as follows. With the Lagrangian experiments presented above, we create a training and a

125    validation dataset using the first simulation setup POLGYR1 (2002 to 2008) and obtain a total of 10260 samples. We divide the samples into a training database (8604 samples from January 2002 to September 2008) and an evaluation database (1224 samples from year 2009). Note that data from September 2008 to January 2009 are not used. The two datasets are separated by a 100-day period to ensure statistical independence. For the test dataset, 6800 samples are created with POLGYR2. They are independant of the training and evaluation experiments.



## 3.2 Architecture

As shown in Figure 4, we use a U-net architecture type (Ronneberger et al., 2015). The U-Net is a state-of-the-art neural architecture for a wide range of image-to-image mapping tasks, including application to ocean studies (Barth et al., 2020; Beauchamp et al., 2022).

The input to the network stacks the hydrodynamic conditions and a $8 \times 100 \times 100$-dimensional tensor representing the initial PDF of the particles. All the initial PDFs have probability of 0.25 over the 4 points in the center of the map, representing the release patch (not shown). The network consists of a series of convolutional (kernel = 5), ReLu and Batch norm layers. A 3-step pooling process allows the resolution to be progressive downscaled from 8 km to 32 km. Combining these steps with skip connections enables the detection of hydrodynamic structures at different spatial scales. The final layer of the U-Net is a softmax layer to provide normalized PDF predictions. The output of the network is a $8 \times 100 \times 100$-dimensional tensor composed of the predicted PDFs from 900 m to 200 m (every 100 m). The last PDF at 200 m represents the predicted origin of the particles.

## 3.3 Training scheme

As training criterion, we consider the Batthacharyya coefficient (Bhattacharyya, 1943) to assess how well the predicted PDFs match the real ones. The Batthacharyya coefficient provides a similarity score between two PDFs, and leads to the Batthacharyya training loss (BL):

$$BL = 1 - BC = 1 - \Sigma \sqrt{P_i Q_i}$$

where $P_i$ is the predicted probability at point i and $Q_i$ is the true probability at the same point. BL ranges from 1 to 0, with 0 representing a perfect prediction. During the training phase, we aim to minimize the following loss function (L) :

$$\mathcal{L} = \frac{BL_{200m} + BL_{300m} + ... + BL_{800m} + BL_{900m}}{n_{layer}}$$

L is the mean of the BL computed at each of the 8 vertical layers. The aim is to force the model to consider the evolution of particles along the water column. Empirically, this method improves the performance of the trained model compared to experiments where the training loss is based only on $BL_{200m}$.

We implement our deep learning scheme using PyTorch (Paszke et al., 2019). For the training phase, the Adam optimization algorithm (Kingma and Ba, 2015) is used with the following hyperparameters: $\beta$ = (0.5, 0.999), no weight decay, and a learning rate of 0.001. The training process is performed using mini-batches of size 32. After 50 training epochs, the best model is selected based on its performance on the evaluation data. We further improve the performance and robustness of the model by using a bootstrapping method with 10 replicates (Breiman, 1996). The final prediction is a set of PDFs computed as the median of the predictions from the 10 models, followed by re-normalization step. We also compute the standard deviation of the 10 predictions as a confidence index (see section 5).





**Figure 4.** General architecture of the Unet. The inputs are temperature, vorticity, u, v, and ssh images at multiple time steps and depths concatenated with initial PDFs (not shown). The outputs are the PDFs at 8 depths representing the funnel of the particles. At the top of each layer we show the resolution of the images and at the bottom the number of channels.



## 4 Results

We report the numerical experiments performed on the considered PAP case study. First, we detail a benchmarking experi-
ment to evaluate the performance of the proposed deep learning schemes. Second, we further analyse how the hydrodynamic
conditions affect the prediction performance.

### 4.1 Prediction performance and comparison

We evaluate the performance of several models on the test database using the $BL_{200m}$ metric (i.e. the Bhattacharyya distance
between the predicted PDF and the true PDF for the catchment area at 200m depth). After a visual analysis of the predictions
and associated scores, we set up the following evaluation criterion. We consider a prediction to be valid if $BL \leq 0.2$, and not
valid if $BL < 0.3$. For predictions with a value of $BL$ between 0.2 and 0.3, the global area is often still well predicted, but the
values of the PDF are usually rough. In these cases, we decide to create a third class called "quasi-valid". These predictions are
not considered valid, but still provide valuable information in the search for particle origins, and this information needs to be
highlighted in the final score statistics.

We benchmark different models in Table 1. The $Unet_{4layers}$ model is the U-net presented in section 3, trained and tested
with the $D_{4layers}$ database, which contains both surface and sub-surface information. The $Unet_{surf}$ model uses only surface
information from the $D_{surf}$ database. $Unet_{surf-nb}$ is similar to $Unet_{surf}$ but without the use of the bootstrap method. The
$CNN_{surf-basic}$ model includes a simple CNN architecture consisting of a series of convolutional layers with a Relu activation
function, which we tested with $D_{surf}$. We also include a $Baseline$ prediction given by the average of all the true PDFs used
in the training database. This mean PDF roughly corresponds to a 2D Gaussian distribution centred on PAP, as also presented
in Wang et al. (2022).

The $Baseline$ and $CNN_{surf-basic}$ models have poor performance with 0 valid cases. This highlights the complexity of the
prediction task and the need for non-trivial models. It is noteworthy that the U-net architecture seems to be well suited to this
problem. $Unet_{surf-nb}$ achieved a very good performance of 58 % of valid cases, which was increased to 66% with the use
of bootstrapping. Taking into account the quasi-valid predictions, $\sim 85\%$ of the predictions provide valuable information with
only surface data. However, the comparison between $Unet_{4layers}$ and $Unet_{surf}$ demonstrates the importance of subsurface
information to obtain more accurate and robust predictions. With 81 % of valid cases and 13 % of quasi-valid cases, $Unet_{4layers}$
clearly outperforms $Unet_{surf}$. This would mean that in about 10-15% of the cases, subsurface data is required to make valid
predictions. Figure 5 shows examples of predictions from the test database using the $Unet_{4layers}$ and $Unet_{surface}$ models. The
PDF is represented by two contours, one containing 99% of the particles (dotted contour) and the other containing 50% of the
particles (solid contour). Both models have a good ability to predict the overall catchment area. However, valid predictions are
mainly characterised by a well predicted center of the PDF, which is usually more accurate for $Unet_{4layers}$. In the following,
we only focus on $Unet_{4layers}$ and $Unet_{surface}$ predictions.





**Figure 5.** Examples of predictions. PDFs predicted with $Unet_{4layers}$ (red) and with $Unet_{surf}$ (green). Black contours represent the true origins of particles. Dotted contours contain 99% of the particles, solid contours contain 50 %. $BL_{200m}$ score is given for both predictions on the top right. On the background : snapshot of SSH at day +20 after the first particle release with the surface velocities represented with the arrows. All domain size are 800 x 800 km and centered on the trap where particles are released.





**Table 1.** Score and performance for different model architectures. For each model we indicate the number of parameters (nb param), the averaged prediction score $BL_{200m}$, the percentage of valid, quasi-valid and non-valid test cases.

| Models | Nb param | $BL_{200m}$ | Valid prediction [%] | Quasi-valid prediction [%] | Non-valid prediction [%] |
|---|---|---|---|---|---|
| $Unet_{4layers}$ | 1.9M (x10) | 0.13 | 81 | 13 | 6 |
| $Unet_{surf}$ | 1.8M (x10) | 0.18 | 66 | 19 | 15 |
| $Unet_{surf-nb}$ | 1.8M | 0.21 | 58 | 23 | 20 |
| $CNN_{surf-basic}$ | 205K | 0.61 | 0 | 1 | 99 |
| $Baseline$ | 0 | 0.45 | 0 | 8 | 92 |

## 4.2 Statistical analyse of the performance

185 We quantitatively analyze potential causes of valid or invalid predictions. The examples show that the shape of the particle distributions has a significant impact on score prediction. We find that PDF far from the ST with mass center > 150 km (i.e the average distance of the particles from their source) are generally not well predicted (Figure 5 and Table 2, index 16, 21, 25). Similarly, complex PDF shapes characterised by multiple patches spread across the domain (Figure 5 index n°2, 15, 16 in ) appear difficult to predict. The entropy ($-\Sigma p_i log(p_i)$, where $p_i$ is the PDF value at point i ) characterises the complexity

190 of the PDF beyond its variance. The entropy increases with the spread of particles and for multimodal distributions and high entropy value ( > 6.6) could explain in some cases the prediction bias (Table 1 index n°2, 15, 16). This trend is confirmed by the statistical analysis over the whole test dataset shown in Figure 6. With a mass center above 200 km, the chance of having a valid prediction with $Unet_{surf}$ is less than 30%. Similarly, we observe a trend of progressive degradation of the score for PDFs with an entropy higher than 6.6 (Figure 6b).

195 Several physical factors could be responsible for the final PDF shape. To characterise the local dynamics, we choose to compute the Mean Kinetic Energy (MKE) and the Eddy Kinetic Energy (EKE) around the ST such as:

$$\underbrace{\frac{1}{2}(\overline{u^2 + v^2})}_{KE} = \underbrace{\frac{1}{2}(\overline{u}^2 + \overline{v}^2)}_{MKE} + \underbrace{\frac{1}{2}(\overline{u'^2 + v'^2})}_{EKE} \tag{1}$$

where u and v are the two components of the horizontal ocean velocity, $\overline{u}$ and $\overline{v}$ are the velocities averaged over a 30-day window, corresponding to the characteristic time of our Lagrangian experiments. Finally, u' and v' are defined as follows:

200 $$u' = u - \overline{u} \tag{2}$$

$$v' = v - \overline{v} \tag{3}$$

MKE and EKE are spatially averaged within a 400 km box around the ST, and vertically between 200 m and 1000 m. These averages are computed every 10 days. MKE can be associated with large-scale currents and mesoscale eddies that stay stable





| Index | $BL^{surf}_{200m}$ | $BL^{4L}_{200m}$ | Mass center | Entropy | MKE | EKE | $\zeta/f_{10^{-2}}$ | $OW/f_{10^{-3}}$ |
|---|---|---|---|---|---|---|---|---|
| 10th per | 0,06 | 0,04 | 36 | 5,7 | 69 | 32 | -4,1 | -1,4 |
| 50th per | 0,15 | 0,11 | 99 | 6,3 | 95 | 49 | -0,4 | 2,2 |
| 90th per | 0,35 | 0,26 | 193 | 6,7 | 126 | 71 | 3,6 | 12,1 |
| 1 | 0,07 | 0,05 | 136 | 6,5 | 90 | 81 | -4,6 | 7,4 |
| 2 | 0,27 | 0,24 | 136 | 6,9 | 94 | 65 | -2,4 | 2,0 |
| 3 | 0,06 | 0,09 | 110 | 6,4 | 101 | 82 | 4,2 | -1,1 |
| 4 | 0,02 | 0,01 | 63 | 5,7 | 124 | 31 | -4,9 | -2,6 |
| 5 | 0,19 | 0,13 | 183 | 6,1 | 127 | 46 | 0,9 | 4,0 |
| 6 | 0,12 | 0,10 | 216 | 6,4 | 107 | 50 | -0,4 | -0,7 |
| 7 | 0,19 | 0,10 | 109 | 6,0 | 96 | 44 | -4,4 | -2,0 |
| 8 | 0,17 | 0,07 | 164 | 5,9 | 98 | 57 | 0,8 | 3,3 |
| 9 | 0,08 | 0,05 | 172 | 6,0 | 82 | 39 | 3,0 | 7,9 |
| 10 | 0,12 | 0,07 | 178 | 6,3 | 79 | 47 | 3,2 | 1,3 |
| 11 | 0,03 | 0,05 | 62 | 6,2 | 89 | 58 | -5,1 | 4,6 |
| 12 | 0,13 | 0,13 | 51 | 6,7 | 65 | 47 | -2,4 | 1,1 |
| 13 | 0,20 | 0,09 | 148 | 6,4 | 74 | 39 | 2,2 | 0,8 |
| 14 | 0,04 | 0,16 | 114 | 5,6 | 84 | 36 | 1,1 | 1,7 |
| 15 | 0,18 | 0,11 | 68 | 6,8 | 114 | 38 | 1,8 | 3,0 |
| 16 | 0,29 | 0,28 | 242 | 6,6 | 91 | 53 | -3,7 | 0,9 |
| 17 | 0,01 | 0,00 | 24 | 5,5 | 131 | 36 | -5,0 | -2,2 |
| 18 | 0,17 | 0,10 | 110 | 5,9 | 60 | 41 | 1,7 | 0,3 |
| 19 | 0,06 | 0,04 | 95 | 6,7 | 137 | 45 | -2,5 | 2,6 |
| 20 | 0,11 | 0,07 | 133 | 6,7 | 85 | 31 | 2,2 | 3,8 |
| 21 | 0,39 | 0,34 | 257 | 6,0 | 89 | 51 | -1,1 | 2,0 |
| 22 | 0,13 | 0,10 | 114 | 6,5 | 89 | 32 | -1,0 | 4,4 |
| 23 | 0,17 | 0,07 | 181 | 6,4 | 98 | 35 | 0,2 | 0,3 |
| 24 | 0,16 | 0,10 | 140 | 6,3 | 132 | 40 | -1,8 | 20,6 |
| 25 | 0,38 | 0,22 | 270 | 5,7 | 92 | 27 | 0,6 | 0,3 |

**Table 2.** Variables associated to the Lagrangian experiments. $BL^{surf}_{200m}$ and $BL^{4L}_{200m}$ indicates the Bhattacharya loss at 2O0m for surface and 4L model. We computed the PDFs' mass center [m] and entropy. The MKE and EKE [$cm^2.s-2$] are computed with a temporal window of 30 days (period of the Lagrangian experiment) and averaged between 200 m and 1000 m in a 400 km box centred at the ST. $\zeta/f$ and $OW/f$ are computed from a surface snapshot taken 20 days after the first particle release and averaged inside a 100 km box centred at the ST. The values of the median, 10th percentile and 90th percentile are given at the top. Index number correspond to the exemples given Figure 5.





during the time window. A large MKE implies strong velocities that are likely to transport the particles far from their source,
further complicating the prediction process. EKE typically indicates the presence of moving mesoscale eddies or submesoscale
fronts. These smaller scale dynamics can promote divergent flows and increase the entropy, thus affecting the prediction score.

The number of invalid predictions increases on average with both MKE and EKE (Figure 6c, d), although the relationship
between the prediction and EKE/MKE is not always simple when looking at all the cases in Table 2. Moreover, the influence of
MKE seems to explain the seasonal patterns we observe with the 5-year temporal coverage (Figure 7). For both $Unet$ models,
the best performance was achieved in winter/early spring, while the highest probability of a non-valid prediction was observed
in summer. We found a strong correlation ($r^2 = 0.64$) between this trend and the seasonal evolution of MKE. This observation
leads us to suggest that MKE could be the main driver of the score on long time scales. The EKE evolution shows a weaker
correlation ($r^2 = 0.13$). In particular, the EKE peak visible in April, associated with the deepest mixed-layer and intensified
submesoscale activity down to 500 m (Buckingham et al., 2016), is not clearly reflected in the performance score.

However, Table 2 suggests that MKE and EKE are not sufficient to explain the prediction score in all cases. The presence of
coherent vorticity structures above the trap appears to facilitate the prediction (Figure 5 index 4, 11, 17). To demonstrate this,
we analyze dynamical features using three indicators: the surface relative vorticity $\zeta = v_x - u_y$, the Okubo-Weiss parameter
OW $= \sigma^2 - \zeta^2$ with $\sigma = (u_x - v_y) + (v_x + u_y)$ and the sea level anomaly $SLA = SSH - <SSH>$ where $<SSH>$ is the
averaged SSH within the subdomain. All these variables are taken from a surface snapshot on the 20th day after the first release
and averaged within a 100 km × 100 km area centered on the ST (Figure 6e, f, g and Table 2). The best performances ($>$
80% of valid predictions) are associated with low negative OW, high absolute $\zeta$ and SLA. This situation generally corresponds
to the presence of a vortex structure above the trap (Wang et al., 2022). On the other hand, the chaotic situations that can be
described by $OW > 0, \zeta \sim 0$, and $SLA \sim 0$ are generally associated with the worst performance.

The presence of large scale currents and eddies appears to be sufficient to explain the prediction scores to a first order
approximation: Large KE is typically associated with lower scores, unless we have coherent structures above the trap (cor-
responding to large KE with negative OW and large vorticity amplitudes). This is highlighted by the bin statistics in Figure
8. The 'non-valid' prediction area in Figure 8a is defined by $KE > 160 \ cm^2.s-2$ and $|\zeta/f| < 0.05$, and is characterized
by an averaged Bhattacharyya score greater than 0.2 (indicated by the red bins). Similarly, it is possible to observe the same
'non-valid' prediction zone in Figure 8b, defined by $KE > 160 \ cm^2.s-2$ and $OW/f > -0.0075$.

## 5 Discussion

In this section, we discuss the possible interpretations of the results, present the limitations of the model, and explore potential
improvements that will guide our future work.

### 5.1 Interpretation of statistics and confidence index

Our results support the relevance of the proposed machine learning approach to predict the catchment area of particles using
only surface data. Although subsurface information can improve the quality of the prediction, it is not essential in most cases.





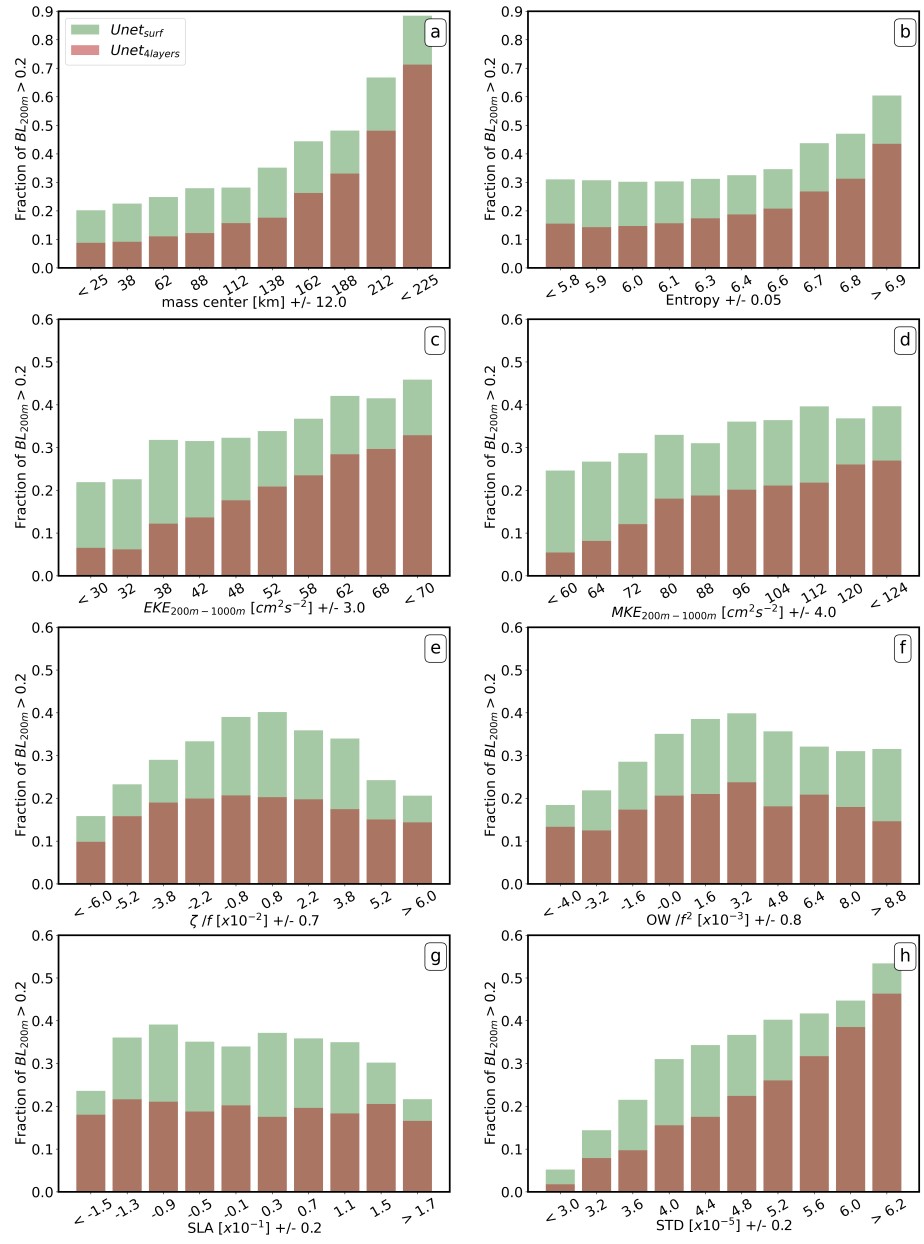

**Figure 6.** Fraction of non-valid prediction ($BL_{200m} > 0.2$) depending on local dynamical variables a) PDF mass center shift, b) PDF Shannon entropy $-\Sigma p \log(p)$, c) Eddy Kinetic Energy (EKE), d) Mean Kinetic Energy (MKE), e) relative surface vorticity $\zeta/f$, g) Surface relative Okubo-Weiss OW/$f^2$, g) Sea Level Anomaly (SLA), and h) standard deviation (STD) of the 10 bootstrap models. For c and d, the energy is averaged in a 400 km box centred on the ST and between 200 and 1000 m. A temporal averaging window of 30 days is used for MKE and EKE. For e, f, g the local dynamical variables come from surface snapshot taken 20 days after the first particle release and averaged in a 100 km box around the ST.



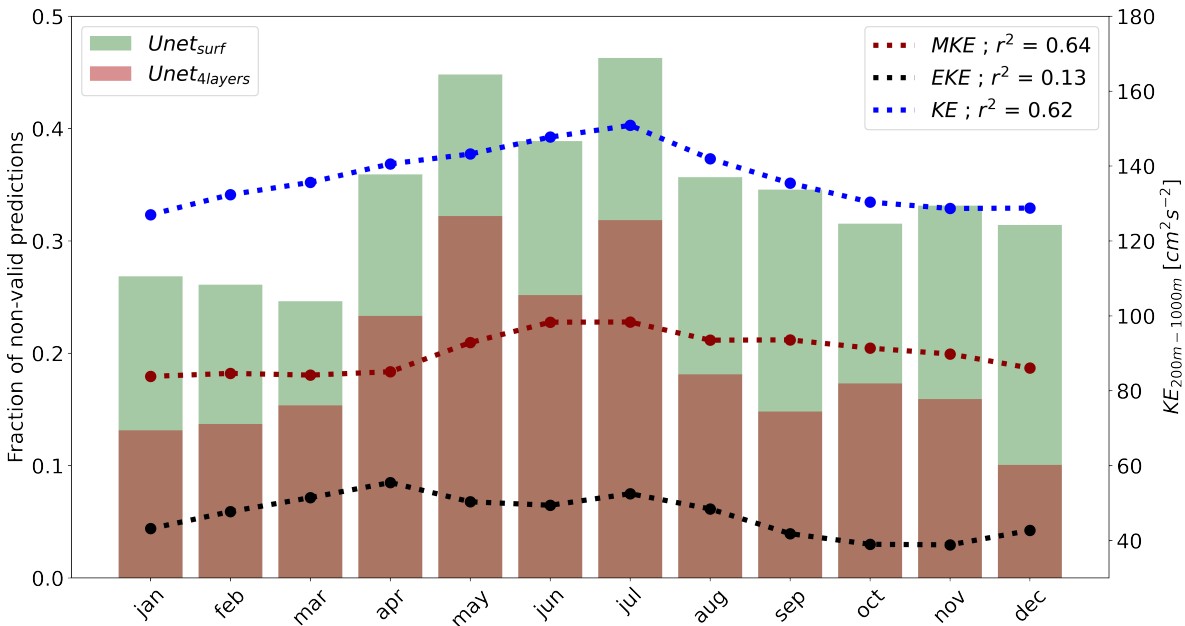

**Figure 7.** Monthly performance statistics over the testing period for $Unet_{surf}$ and $Unet_{4layer}$. Bar score indicate the probability of non-valid prediction ($BL_{200m} > 0.2$). The monthly averaged evolution of EKE (dotted) and MKE (dashed) between 200 and 1000 m is shown in black.

Analysis of the drivers of the prediction score revealed complex multifactorial causes. Only a few indicators have been chosen for this analysis, and we do not exclude the possibility of other links that have not been investigated. However, the statistics appear coherent to us, as they suggest that the model is more robust under weak and/or stable dynamics. Given that our model is limited in time and space by the input resolution (8 km resolution, only surface or 4 vertical layers and snapshots
every 10 days), it is clear that the predictions are sensitive to finer spatial and temporal variability. This may explain the good performance with small KE or in the presence of coherent eddies above the trap. Eddy structures can have a clear surface signature that is easy to identify and that stay coherent at depth (Figure 2). Considering a 30-day period, they are generally stable showing little temporal variability, limiting the particles' spatial dispersion trapped inside. In contrast, chaotic situations, typically characterized by randomly distributed submesoscale fronts with a horizontal size of about 10 km, are more likely to
be less well predicted. Additionally, such structures can be too small to be detected. Temporal evolution can be very fast (on the order of days) and is unlikely to be adequately captured by our machine learning models. Such assumptions can directly help in the future to develop a confidence index that depends on the local dynamics. For future real applications, it will probably be difficult to predict the catchment area in every situation. The aim is therefore to understand in which situation the predictions made by the machine learning model can be trusted. In this respect, the standard deviation (STD) of the 10 model predictions





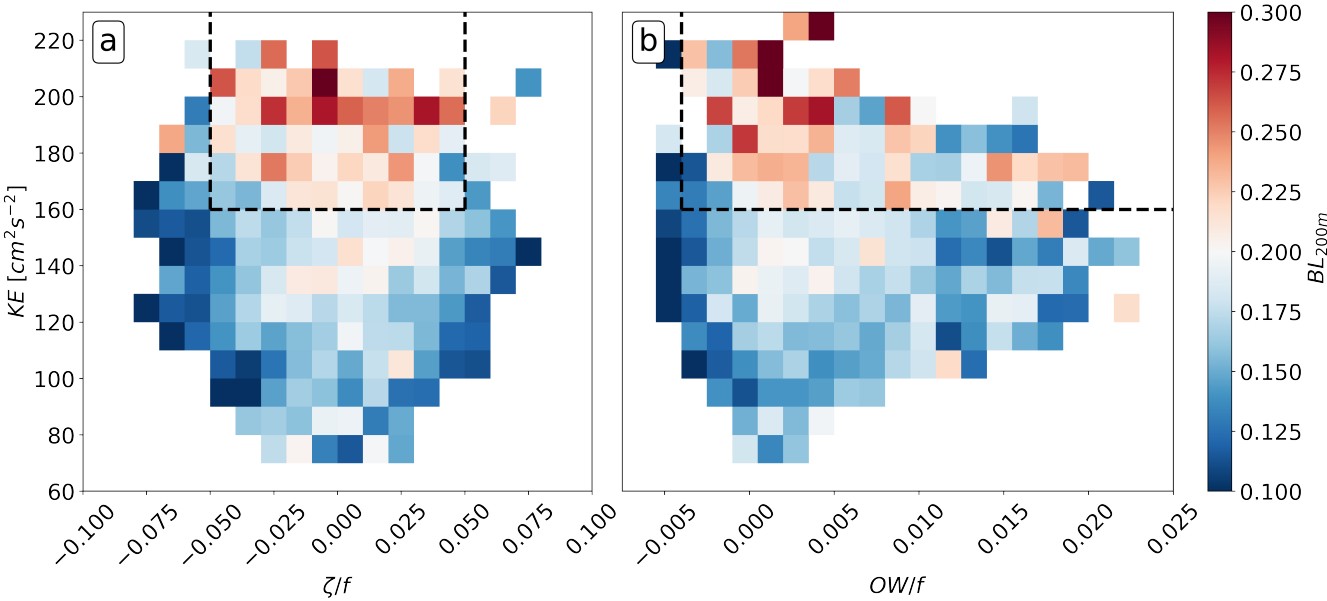

**Figure 8.** Averaged bin statistics of the prediction score in the a) Kinetic Energy - relative vorticiy space b) Kinetic Energy - relative Okubo-Weiss space. Color represents the mean Bhattacharyya score. Bins with less than 5 samples inside are masked. The dashed black lines delimite the "non-valid" prediction zone represented by the red bins.

from the bootstrap method (Figure 6f) can also provide insight into quantifying the uncertainties of the predictions. The STD, computed as the mean STD of the 10 models at each prediction point, shows a linear relationship with the probability of having a valid prediction for the Unet models. Previous studies have also explored such bootstrapping methods for uncertainty quantification (Pauthenet et al., 2022; Beauchamp et al., 2022; Haynes et al., 2023). Future work could also explore more advanced deep learning schemes with built-in uncertainty quantification properties (Haynes et al., 2023).

**5.2    Towards real data and a more realistic representation of particles**

The results strongly suggest a possible use of satellite data to more precisely identify the source area of the particles measured in the deep ocean. As emphasized by Fevre et al. (in prep) for the learning-based mapping of sea surface dynamics, the representativeness of the numerical simulations w.r.t. the real 3D +T hydrodynamic conditions is likely to be a critical feature for the application of a neural model trained from simulation data to a real-world configuration. The use of ocean reanalyses

(Frigstad et al., 2015) could also be a relevant solution. In both cases, an in-depth analysis of the 3D ocean circulation seems particularly important to assess the potential limitations of the proposed scheme.

Regarding biology, we consider a ST at 1000 m, whereas in reality the ST at PAP are moored deeper ($\sim$ 3000 m). It has been shown that below 1000 m, currents are weaker and do not significantly affect the horizontal displacement of particles (Wang et al., 2022). Therefore, although the influence of ST depth on the prediction score needs to be investigated, it can be





assumed that it will not significantly affect the model performance. However, a major bias in the model probably comes from the carbon particle representation. We represent sinking particles with a settling velocity of 50 m/day, whereas particles have a wide range of velocities that can vary significantly seasonally (Villa-Alfageme et al., 2016). The origin of particles can be very different depending on their sinking velocity, so it is important to consider the whole particle velocity spectrum to represent the catchment areas for particles with different velocities (Wekerle et al., 2018). Therefore, the next step of this study is to focus on

the effect of sinking velocities on the prediction score. We can assume that the prediction of catchment areas for particles with higher speeds than 50 m/day is likely to be more efficient, as their trajectories are less impacted by currents and turbulence (Liu et al., 2018; Wekerle et al., 2018; Ma et al., 2021; Wang et al., 2022). However, particles with lower sinking velocities are more sensitive to the flow and can easily be dispersed over a large area far away from the ST, which would decrease the prediction score, as suggested in Figures 6b and d.

The use of a coupled biogeochemical model embedded in POLGYR would provide information on surface chlorophyll distribution (i.e. sea colour) and primary production intensity (PP) (Kostadinov et al., 2009; Dunne et al., 2005). Although the relationships are not direct (Laurenceau-Cornec et al., 2020; Iversen and Lampitt, 2020; Cael et al., 2021), high PP levels are associated with large phytoplankton cells and particles, which may be associated with higher sink rates. It is then possible to weigh the distribution of particles in terms of size and sinking velocity against the PP estimated by the model where they

are exported, which should make the simulations more realistic. In addition, sea colour observations, together with altimetry data, can be used as input to a machine learning system to better constrain carbon fluxes at depth, using an approach similar to that presented here. However, particle size and sinking velocity are affected by numerous biological processes during their journey, which are generally not taken into account in simple Lagrangian studies. Although a Lagrangian approach can be developed (Jokulsdottir and Archer, 2016), an ad hoc parameterisation, for example derived from BGC models simulating

particle dynamics (Aumont et al., 2015), could also be used and tested.

## 6 Conclusions

Identifying the origin of particles captured by sediment traps is important for interpreting measured fluxes and improving sampling methods. This task is challenging and often leads to inaccurate assumptions or requires complex 3D data assimilation models. In our study, (i) we demonstrate the ability of machine learning to predict the origin of particles trapped in a PAP

sediment trap in real time, based on realistic 3D numerical simulation and Lagrangian tracking. (ii) We found that the catchment area of particles is mainly driven by ocean surface conditions, suggesting that our machine learning model could be applied to satellite data. (iii) Statistical analysis shows that the performance of the prediction is sensitive to local dynamics. The model performs better at low KE and in the presence of coherent vortices above the sediment trap. The next challenge is to apply our model to real data and develop a confidence index based on the local conditions. The particle modeling also needs to be

improved to account for the wide range of particle sinking velocities.



*Code availability.* The codes used in this study are available online at https://github.com/TheoPcrd/SPARO.

*Data availability.* The datasets used in this study are available online at https://doi.org/10.17882/97556.

*Author contributions.* TP, JG, RF and LM : writing, analyse and methodology. JC : writing and technical support for the simulations.

*Competing interests.* The authors declare no competing interests.

*Acknowledgements.* T.P. received a Ph.D. grant from École Normale Superieure Paris-Saclay. This work was also supported by the ISblue project, an Interdisciplinary graduate school for the blue planet (ANR-17-EURE-0015), and co-funded by a grant from the French government under the program "Investissements d'Avenir". This manuscript contributes to the APERO project funded by the National Research Agency under the grant APERO [grant number ANR ANR-21-CE01-0027] and by the French LEFE-Cyber program. Simulations were performed using HPC resources from GENCI-TGCC (grant 2018-A0050107638) and from DATARMOR of "Pôle de Calcul Intensif pour la Mer" at
Ifremer, Brest, France. J.G. would like to acknowledge support from the French National Agency for Research (ANR) through the project DEEPER (ANR-19-CE01-0002-01) and AI chair OceaniX (ANR-19-CHIA-0016). Simulations were performed using HPC resources from GENCI-TGCC (Grants 2022-A0090112051), and from HPC facilities DATARMOR of "Pôle de Calcul Intensif pour la Mer" at Ifremer Brest France. The authors thank Mathieu Le Corre for providing CROCO simulation outputs.



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
