# Peer review of "Learning-based prediction of the particles catchment area of deep ocean sediment traps"

_EGUsphere, 2023_

## Author Comment (AC2)

**Detail response for reviews "Learning-based prediction of the particles catchment area of deep ocean sediment traps"**

Dear editor and reviewers,

We are truly grateful for the time and effort you dedicated to carefully assessing our manuscript. Please find attached our revised manuscript, incorporating the revisions based on your feedback. We sincerely hope that you find this updated version to be reflective of your insightful suggestions. Please note that the line numbers mentioned here refer to the track changes file.

*general comments*

*This paper presents a novel method for estimating the near surface origin of sinking particles that reach a sediment trap at 1000m. A neural network based on the U-Net architecture is trained to predict backward trajectories of sinking particles from environmental variables. The authors present the method and demonstrate its skill in a twin experiment setting. They also provide compelling analysis of the behavior of their estimation method depending on dynamical situation.*

*I feel this is a strong paper and recommend it to be accepted for publication after minor revision. The paper is generally very well written and clear. The method being introduced opens up interesting perspectives to help better interpret sediment trap observations, but the authors also acknowledge difficulties that lie ahead if such method was to be applied to real data. One of the strengths of this paper in my view is the analysis of results in relation to physical processes. A weakness is that we don't know how/if the method's skill could be quantified in the real world.*

*specific comments*

*- L66 : please provide a reference to justify the choice of "50m/day" as realistic on average in the real world, assuming that this indeed is the case. Is 50m/day representative of a large fraction of sinking biomass that sediment traps capture? Wang et al 2022, cited later, appears to be a modeling study by the same authors. Please state in the paper whether the choice of 50m/day is backed by real world observations, or otherwise.*

As mentioned in the discussion, representing only one sinking speed is a significant limitation that will be evaluated in a subsequent study. However, as a preliminary step, we have chosen 50 m/day, which represents the slow range of sinking particles observed in the region at depth (Villa-Alfageme et al., 2014). Slow particles are more susceptible to being affected by ocean circulation, which in turn makes the prediction of catchment areas more challenging. We revised the manuscript to better express this choice lines 69-72 :

"*We focus on particles with a vertical sinking velocity of 50 m/day, which represents the slow range of sinking particles observed in the region (Villa-Alfageme et al. 2014). Slow particles*

*are more susceptible to being influenced by ocean circulation, which in turn makes the prediction of catchment areas more challenging.*"

*- L107 : "based on the statistical results of Wang et al 2022" is vague. What statistical results?*

We have revised the sentence lines 116-118:

"*Based on Wang et al 2022, who computed the source region at 200 m for particles collected by the moored sediment traps over the seven years (2002–2008), an horizontal domain of 800 km x 800 km centered on the ST was chosen to encompass all the source particles position.*"

*- L138 : please explain and/or provide a reference for "Combining these steps with skip connections enables the detection of hydrodynamic structures at different spatial scales".*

*- Fig 4 : more explanation about "concatenation" and "channels" would also be useful in section 5.2 for the non specialist reader.*

We have revised the paragraph. We have added Ronneberger et al. 2015 reference to justify that "skip connections enable the detection of hydrodynamic structures at different spatial scales" and we have detailed the meaning of the terms "channel" and "concatenation" :

" *A 3-step pooling process is used to downsample the channel, i.e. the dimension representing the number of dynamical images for one experiment. At each step, this number of features is doubled. This successive resolution downscaling from 8 km to 32 km, combined with skip connections and concatenation (addition over the channel dimension), could facilitate the detection of structures at different spatial scales (Ronneberger et al. 2015).*"

*- L178 : can it be excluded that Unet4layers simlply benefits from a larger number of data constraints (Nx) that in UnetSurf rather than their subsurface location? I wonder if providing Nx time as many surface data to train UnetSurf would make it match the performance of Unet4layers.*

The number of images taken as an input ($N_{inputs}$) shouldn't affect the score performance.

We tested this by increasing the number of channels in the first layer (128 instead of 64 channels), but this had no impact on the final score. In other words, we believe that the predictions are not limited by the number of parameters in the Unet but by the lack of information required in the inputs.

*- L220-223 : state how much of the training sets was "chaotic situations" versus nonchaotic ones. I wonder : if you trained a NN separately on just "chaotic situations", would you get improved performance? maybe match the performance obtained for nonchaotic ones.*

According to our criterion based on vorticity and KE, there are 1818 chaotic cases (27% of the experiences in the testing dataset). We have added this information in the manuscript line 269 :

"This "chaotic situation" represents 27% of our dataset."

Since the number of chaotic cases is significant, it would indeed be interesting to train specific NN depending on the dynamical situation and this will be investigated in a future study.

*- L294 : it seems to me that a confidence index would need to correctly account for several sources of model error (uncertain sinking velocity distributions, possible biases in POLGYR statistics, transport rates, etc) to avoid being misleading. Please discuss this in a bit more detail.*

Indeed, the uncertainties associated with the numerical simulation must also be taken into account when calculating an index of confidence. We have mentioned this in the discussion lines 296-299 :

"*However, it is worth noting that other sources of uncertainty, such as sinking velocity distributions, simplification of numerical simulations, transport rates, etc., would also be important to evaluate.* "

*- in section 5 please describe the kind of real world observational experiment and data sets that would be needed to demonstrate / quantify the method's skill outside of a twin experiment configuration. I feel that many would be concerned with using real world results of such a method if its skill can only be assessed within a model world.*

A possible strategy to validate the model predictions with real observations would be to evaluate the cross-correlation between satellite chlorophyll in the prediction area and carbon fluxes measured at PAP (Frigstad et al. 2015). The cross-correlation coefficient can be compared with the cross-correlation obtained by considering a simplified catchment area, such as a 100 km or 200 km box around the PAP location, which is classically still used today (Lampitt et al. 2023). If a better correlation is found, this would indicate a better connection between deep fluxes at PAP and surface images from satellites and validate the relevance of the model to real observations.

We have added this details lines 301-306:

"*A possible strategy to validate the model predictions with real observations would be to evaluate the cross-correlation between satellite chlorophyll in the prediction area and carbon fluxes measured at PAP (Frigstad et al. 2015). The cross-correlation coefficient can be compared with the cross-correlation obtained by considering a simplified catchment area, such as a 100 km or 200 km box around the PAP location, which is a classical method still used today (Lampitt et al. 2023). If a better correlation is found, this would indicate a better connection between deep fluxes at PAP and surface images from satellites and confirm the relevance of the model application to real observations.*"

*technical corrections*

*L57 : "used to design"*

The correction has been applied.

*L94 : space missing in "1000 m.Biological"*

The correction has been applied.

*L96 : spell out what 3D+t means. also, consider saying 4D instead of 3D+t or 3D+T*

We gave a more explicit definition of "3D+T" line 103 :

"*The PDFs saved at vertical levels lower than 200 m provide information on the 3 spatial dimensions and temporal dimension (3D+T)*"

*L100 : "To avoid common particle between two experiments" is unclear and seems grammatically incorrect*

We removed the sentence and we added in the "Training, validation and test data" section lines 138-143 :

"*With the Lagrangian experiments presented above, we create a training and a validation dataset using the first simulation setup POLGYR1 (2002 to 2008). During this period, Lagrangian experiments are realised every 10 days time step and at 36 fictive ST positions around the PAP region resulting in a total of 10260 samples (Figure 3).*"

*L101 : rephrase as e.g. "to set up the patch centers 36km apart"*

The correction has been applied.

*L104 : nx,ny would seem a better notation than deltax, deltay; no?*

The correction has been applied.

*L109 : "for storage constrain" is vague and seems grammatically incorrect.*

We have removed the term.

*Fig 3 caption : "superimposed" (one "s" only)*

The correction has been applied.

---

## Author Comment (AC3)

**Detail response for reviews "Learning-based prediction of the particles catchment area of deep ocean sediment traps"**

Dear editor and reviewers,

We are truly grateful for the time and effort you dedicated to carefully assessing our manuscript. Please find attached our revised manuscript, incorporating the revisions based on your feedback. We sincerely hope that you find this updated version to be reflective of your insightful suggestions. Please note that the line numbers mentioned here refer to the track changes file.

*The study is heavily based on the Lagrangian simulations of particle sinking based on circulation model results and described in Wang et al., 2022 (I think this should be stated more explicitly and include a brief summary of results. Best in Section 2). The authors use the results of these simulations to test the possibilities of using machine learning to locate the origin of particles caught in deep sea sediment traps. The aim is to use observed sea surface conditions (mainly satellite altimetry and SST) to deduce the source of particles.*

We mentioned Wang et al. 2022 methodology at the beginning of section 2 :

"*The strategy is based on Wang et al 2022, who characterised the origins of particles collected at PAP using a simulation-based experimental setup consisting of realistic CROCO simulations and particle backtracking experiments.*"

*The authors demonstrate that the machine learning algorithm vastly outperforms the baseline prediction, which uses the average distribution centred above the sediment trap. However, they should give a more detailed explanation of what are the benefits of using their algorithm over running the Lagrangian simulations. To put it in another way, why not simulate the particle paths using existing ocean reanalyses instead of 'guessing' the paths from surface observations? There are many regional model reanalyses of ocean currents available and e.g. CMEMS offers global reanalysis in roughly 8 km resolution which matches the one used in this study. It is clear that there are computational benefits, but one would think that with all the efforts invested in setting the sediment traps, dedicating some serious computing time shouldn't be a problem. The answer might be obvious to the authors, but certainly not to the potential audience. The authors state in the introduction that there are uncertainties stemming from the ocean model errors. That is certainly true, but in such case I am missing at least a rough comparison of the accuracy of their method to the uncertainties in the prediction of the catchment areas using the Lagrangian backtracking.*

*I am not a native speaker, so my comments on the English grammar are merely suggestions.*

The use of backtracking in reanalysis is indeed an attractive approach that can be used to predict the surface source of particles. However, ocean reanalyses are likely to have significant uncertainties in the mesoscale-to-submesoscale range, which seems to be key for

the backtracking of catchment areas. Despite the 8 km grid resolution of ocean reanalyses, the smallest horizontal scales resolved by observation-driven and model-based products for sea surface dynamics are much coarser. They are typically below the scale of satellite altimetry (Fevre et al. 2023, table 2), which is about 100 km in our region (Ballarota et al. 2019). In addition, there are significant uncertainties associated with the dynamics computed in the subsurface, in particular the vertical structure of mesoscale-to-submesoscale structures, which play a critical role in particle trajectories. Thus, backtracking with reanalysis datasets is associated with significant uncertainties, which are difficult to assess due to the lack of synoptic-scale data. We plan to evaluate these aspects more thoroughly in a future study.

Using machine learning to predict the location of particle sources simplifies the experimental setup and saves computational resources: once trained, the model can be applied directly to any period of available surface data, including the most recent one, without any backtracking simulation. By estimating the catchment area directly from satellite data, we also avoid the potential bias associated with reanalyses in the ocean's interior. Moreover, in contrast to reanalysis, we show in this study that our estimate can be associated with a confidence index that can depend on either local dynamical conditions (fig 8) or bootstrap model variability (fig 6. h.). Finally, to the best of our knowledge, no study has investigated how ocean surface conditions constrain particle trajectories. We believe that the comparison we made between the 4-layer and surface-only Unets provides new and significant information. In particular, it shows that in most cases surface information is sufficient to have a good estimate of the particle sources. Although hoped for, this result was not anticipated and was by no means obvious. Hence, we believe that this new approach can be relevant in reaching this conclusion.

To better address these points, we have revised :

- The introduction lines 50-54 :

"*In addition, while it can be complex to investigate how the surface constrains particle motion at depth using 3D reanalysis, machine learning offers new ways of assessing the main drivers of particle displacement. This can lead to a better understanding of the variables and processes involved in particle sinking, and a clearer identification of the temporal/spatial resolution required for effective particle trajectory reconstruction.*"

- The conclusion :

"*Identifying the origin of particles captured by sediment traps is important for interpreting measured fluxes and improving sampling methods. While it remains complex to investigate how the surface constrains particles' motion at depth with 3D reanalyses, we exploit the proposed machine learning scheme to address this question. In our study, (i) we demonstrate the ability of machine learning to predict, in real time, the origin of particles trapped in a PAP sediment trap at 1000 m depth based on realistic 3D numerical simulation and Lagrangian tracking. (ii) Using sensitivity analyses with different input data, including varying their space-time resolution, our experiments support a sea-surface-only configuration to relevantly predict particles' catchment area in most cases, suggesting that our machine learning model could be applied to satellite data. (iii) Statistical analysis shows that the prediction performance is sensitive to local dynamics. The model performs better at low KE and in the presence of coherent surface vortices above the sediment trap. The next challenge is to apply our model to real data and develop a confidence index based on the local conditions. The particle modelling also needs to be improved to account for the wide range of particle sinking velocities.*"

*Comments by lines:*

*L. 7: "surface conditions appear to be sufficient to accurately predict the source area" – Are they? What is considered sufficient?*

The statement is based on the metric we developed. We obtained 85% of valid and quasi-valid predictions with just the surface, while the baseline obtained 8%. We changed the term "sufficient" to "provide valuable information" which is more accurate (line 7) :

 "*Our numerical experiments support its predictive performance, and surface conditions appear to provide valuable information to accurately predict the source area*"

*L. 66: Why 50 m/day?*

As mentioned in the discussion, representing only one sinking speed is a significant limitation that will be evaluated in a subsequent study. However, as a preliminary step, we have chosen 50 m/day, which represents the slow range of sinking particles observed at depth in the region (Villa-Alfageme et al., 2014). Slow particles are more susceptible to being affected by ocean circulation, which in turn makes predicting catchment areas more challenging. We have revised the manuscript to better express this choice line 69-72 :

"*We focus on particles with a vertical sinking velocity of 50 m/day, which represents the slow range of sinking particles observed at depth in the region (Villa-Alfageme et al. 2014). Slow particles are more susceptible to being influenced by ocean circulation, which in turn makes predicting catchment areas under these conditions more challenging.*"

*L. 76: "After spin-up, the chaotic evolution results in uncorrelated dynamics" – How are the initial and boundary conditions perturbed? Shouldn't the same atmospheric conditions ensure similarity of both runs? Large distance from the boundary and same atmospheric forcing would imply very similar oceanographic conditions. This is very important. If the test conditions are related to the training conditions, the performance of the ML algorithm is overrated. This could also (at least partially) explain the results which are better "under weak/or stable dynamics". Such conditions are likely also the situations when POLGYR1 and POLGYR2 currents would be most similar. This should be checked and thoroughly explained in the text.*

The 2 runs use different initial and boundary conditions derived from parent North Atlantic runs with slightly different options/configurations. The most significant of these is a surface salinity restoring, which was added in the second version of the runs but not in the first, resulting in a slightly biased large-scale stratification in the region of interest. The freshwater forcing in the nests is also slightly different due to this restoring. Since we use the same atmospheric  data for the forcings, POLGYR1 and POLGYR2 have similar evolution and statistics at large scales in terms of energy and variability. In contrast, the mesoscale and

submesoscale structures, which are the main cause of particle displacement in our study, are not similar for the same given date due to their chaotic evolution (e.g. Zanna et al, 18).

To illustrate this, we have computed the correlation evolution between POLGYR1 and POLGYR2 with vorticity and SSH for the year 2005 (Figure 1 below). We compare the results obtained with the correlation evolution of the same variables in POLGYR1, but between the years 2005 and 2006. If POLGYR1 and POLGYR2 were dependent, we would observe a higher correlation with the red line compared to the green line, but we do not observea significant difference here.

[Figure]

Figure 1 : Time series of the correlation coefficient between SSH from POLGYR1 and POLGYR2 over the year 2005 (red line) and between SSH from POLGYR1 over 2005 and 2006 (green line). A similar plot is shown with vorticity (blue and orange lines).

Another example can be seen in this snapshot taken on July 24th 2023 (Figure 2). The eddies and frontal structures do not match, resulting in very different particle source areas. Therefore, this experimental setup ensures that the score obtained with the test dataset is not biased by a dependence between the training and test databases.

[Figure]

Figure 2 : Snapshot of the relative vorticity in the PAP area on July 24[th] 2023 for POLGYR1 (left) and POLGYR2 (right).

We have revised the paper by adding the last figure in the annex and mentioning it in line 83 (Figure A1).

*L. 90: Why 10 km x 10 km patch? This paragraph is a bit confusing. 36 particles are released from 10 km x 10 km patches, but there are 36 STs in the model and they are 36 km apart. Is this right? So 36x36=1296 particles are released every 12 hours? Why not release them from 36 points instead of patches? To compensate for the lack of dispersion? The paragraph should be rewritten and clearer.*

This is correct. We chose a 10 km patch to account for the dispersion not represented by a Lagrangian approach, and a total of 1296 particles are released every 12 hours. We have revised the paper by introducing the 36 STs position in the section 3.1 (Training, validation and test data) instead of the 2.2 (Lagrangian experiment). This appears more coherent to us because section 2.2 only explains the methodology for 1 Lagrangian experiment and 3.1 section explains where we apply these experiments and at what frequency (i.e. every 10 days and at 36 different positions over 8 years).

Revised paragraphe lines 136-143 :

"*Following the training and evaluation frameworks used in deep learning studies (Lecun et al., 2015), we consider independent training, validation and test datasets as follows. With the Lagrangian experiments presented above, we create a training and a validation dataset using the first simulation setup POLGYR1 (2002 - 2009). During this period, Lagrangian experiments are realised every 10 days time step and at 36 fictive ST positions around the PAP region to increase the number of experiments (Figure 3a ). The STs are close enough to the PAP site to have the same hydrodynamic properties. We choose to place the patch centers 36 km apart, so that particles from two different patches are separated by at least 26 km, which is slightly above the Rossby radius value in the region (Chelton et al., 1998). This*

*distance is sufficient to observe significant differences in the catchment areas for two consecutive patches (Figure 3b), resulting in a total of 10260 samples. "*

*L. 104: I think the delta sign is not a good choice for the number of points as it implies the distance between points.*

We have replaced $\delta y$ and $\delta x$ by $n_y$ and $n_x$ .

*L. 108: The authors should explain better the 8 km resolution. I think that the simulations run at 2 km resolution, but the end results are downscaled and so are the surface fields used for the machine learning. This would also match roughly with the resolution of the satellite altimetry. Am I right? If so, this should be clearer from the text.*

Given the limitations of our computational resources, a 2 km resolution dataset proved to be overly challenging. Furthermore,  the effective resolution of the model is expected to be around 8 km (Soufflet et al, 16 https://doi.org/10.1016/j.ocemod.2015.12.004), so we downscaled all data resolution to 8 km We have included a sentence to explain this decision line 120 :

 "Due to computational constraints, and the effective resolution of the model expected to be around 8 km (Soufflet et al, 16), all data horizontal resolution is downscaled to 8 km, resulting in ny = nx = 100 points"

*Figure 3 caption: remove "the" from "the origins". "from the left panel" instead of "of the left panel."*

The correction has been applied.

*L. 142 (something wrong with the line numbering here): "the training criterion". Batthacharyya coefficient is used in the equation as BC and should be marked as such. The sum (epsilon) in the equation is too small and is missing the summation index.*

The equations for BC and for the training criterion have been incorporated, as have the summation index and the index z corresponding to the depth of the PDF (lines 165-175).

*"As training criterion, we consider the Batthacharyya coefficient (Bhattacharyya, 1943) to assess how well the predicted PDFs match the real ones.*

*BCz = Σi∈D √Pi,z Qi,z (1)*

*where D represents the PAP domain, Pi is the predicted probability and Qi is the true probability at point i and at depth z. The  Batthacharyya coefficient provides a similarity score between two PDFs, and leads to the Batthacharyya training loss (BL):*

*BLz = 1 − BCz (2)"*

*L. 145: "Empirically, this method improves the performance of the trained model compared to experiments where the training loss is based only on BL$_{200m}$." - This is very interesting. On one hand, one would expect only the end location to matter, on the other hand, the path is important. It would be interesting to know more.*

It can be observed that the score is indeed improved when the model is trained to reconstruct the entire path of the particles over the water column. However, the score improvement is not particularly significant when compared to a training loss based solely on $BL_{200m}$. Consequently, we did not investigate these results further.

It should be noted that the rationale for training with BL instead of $BL_{200m}$ was also to predict the entire particle funnel over the water column. Such predictions, when coupled with in-situ data such as BGC Argo or gliders, may be worthy of further investigation.

*L. 160: The values of BL seem a bit arbitrary. A visual analysis is kind of a weak argument. What does it mean in the practical sense? Is it possible to relate how much would a certain value of BL affect the content of the sediment trap? L. 184: "analysis" instead of "analyse".*

BL is a classical distance used in data science, with a range between 0 and 1 (i.e Tang et al. 2022 10.1109/LGRS.2022.3161931).  In practice, BL can also be an indicator of the percentage of PDF predicted $\Sigma_{i \in D} min( P_{i,200m} Q_{i,200m} )$ (see the relationship in the figure below). A BL value of 0.3 indicates that approximately 45% of the PDF has been correctly predicted, which corresponds to the percentage of particles that have been correctly identified. Table 1 indicates that, on average, $BL_{baseline} = 0.45$, corresponding to $\sim 30\%$ of the PDF predicted. In contrast, $BL_{surf} = 0.18$ and $BL_{4L} = 0.13$, correspond to an average of respectively $\sim 58\% - 65\%$ of the PDF predicted, which is about twice as much as the baseline.

[Figure]

Figure 3 : Scatter plot of the Bhattacharyya score BL200m vs the percentage of particles predicted, defined as $\Sigma i \in D$ min(Pi,200mQi,200m). The 4D polynomial relationship is shown with a black line. The dashed vertical line represents the limit of the valid, quasi-valid, and non-valid zones.

The figure has been included in the annex (Figure A2) and the paragraph 4.1 (lines 194-198) has been revised as follows:

"*Empirically, we observe a polynomial relationship between the BL score and the percentage of the PDF predicted defined as $\Sigma_{i \in D} min( P_{i,200m} \ Q_{i,200m} )$ (Annex figure A2). We set up the following arbitrary evaluation criterion. A prediction is considered valid if BL is less than 0.2, and invalid if BL is greater than 0.3. These two thresholds correspond to a percentage of the predicted PDF greater than or equal to 55% and 45%, respectively.*"

The correction has been applied.

*L. 219: Why the 20[th] day? Shouldn't these be averaged over the average travel time?*

The particles reach the depth of 200 m approximately between day - 15 and - 25 before the initial release. Consequently, the 20th day represents the oceanic conditions when the majority of particles are situated near to the surface and under the influence of surface

currents. Also, in order to avoid any smoothing effect, a surface snapshot was chosen rather than an average.

We revised the manuscript as follows lines 254-280 :

"*In order to avoid any smoothing effect, all these variables are taken from a surface snapshot taken 20 days before the first particle release, rather than an average over the experience period. The particles reach the depth 200m approximately between day -15 and -25 after the initial release. Consequently, the 20th day represents the oceanic conditions when the majority of particles are situated near to the surface and under the influence of surface currents.*"

*L. 256: Maybe "collected" instead of "measured".*

The correction has been applied.

*L. 275-285: The biogeochemical model results and satellite chlorophyll concentration measurements show the spatial variability of phytoplankton biomass. Maybe this could serve as a measure of needed accuracy of the method? On the other hand, this paragraph focuses on primary production only and neglects other sources of particles such as zooplankton. The latter could be obtained from the biogeochemical model as well.*

Indeed, the spatial variability of the phytoplankton biomass would be a relevant metric to understand the required accuracy of the method, and this issue will be addressed in a future paper. We planned to study chlorophyll as a first step, since we can obtain the data from direct satellite imagery. However, in the future we expect to analyse other particle sources that we can obtain from models such as NNP or zooplankton.

*L. 298: "analysis" instead of "analyse".*

The correction has been applied.